# Multiscale Numerical and Experimental Analysis of Tribological Performance of GO Coating on Steel Substrates

**DOI:** 10.3390/ma13010041

**Published:** 2019-12-20

**Authors:** Robin Hildyard, Mahdi Mohammadpour, Sina Saremi-Yarahmadi, Manuela Pacella

**Affiliations:** 1Wolfson School of Mechanical Electrical and Manufacturing Engineering, Loughborough University, Loughborough, Leicestershire LE11 3TU, UK; M.Mohammad-Pour@lboro.ac.uk (M.M.); M.Pacella@lboro.ac.uk (M.P.); 2Department of Materials, Loughborough University, Leicestershire LE11 3TU, UK; S.Saremi@lboro.ac.uk

**Keywords:** nano-scale friction, atomic force microscopy, thin film, macroscale tribological coatings, graphene oxide

## Abstract

Herein, nano-tribological behaviour of graphene oxide (GO) coatings is evaluated by a combination of nanoscale frictional performance and adhesion, as well as macroscale numerical modelling. A suite of characterisation techniques including atomic force microscopy (AFM) and optical interferometry are used to characterise the coatings at the asperity level. Numerical modelling is employed to consider the effectiveness of the coatings at the conjunction level. The macroscale numerical model reveals suitable deposition conditions for superior GO coatings, as confirmed by the lowest measured friction values. The proposed macroscale numerical model is developed considering both the surface shear strength of asperities of coatings obtained from AFM and the resultant morphology of the depositions obtained from surface measurements. Such a multi-scale approach, comprising numerical and experimental methods to investigate the tribological behaviour of GO tribological films has not been reported hitherto and can be applied to real-world macroscale applications such as the piston ring/cylinder liner conjunction within the modern internal combustion engine.

## 1. Introduction

Frictional losses in most mechanical systems are undesirable. For example, in the internal combustion engine 48% of the energy losses can be attributed to some manner of friction, such as piston skirt friction, piston ring friction and friction within the bearings [1]. One method to reduce these losses is to apply lubricants to the contact experiencing undesirable friction, which creates a high-shear layer between the contacting surfaces, hence allowing the surfaces of the contact to readily slide over each other with decreased resistance. Historically, liquid lubricants have dominated in this application, but in recent decades solid lubricants have been proving their worth and are the focus of significant developments [2,3]. Nevertheless, a lack of fundamental understanding of the role of thin film solid lubricants has resulted in their limited integration and utilisation at the industrial scale despite their attractive tribological [4] and mechanical properties, such as high hardness and high elastic modulus [5].

Graphene, and its derivatives, have been identified as key contributors to the field of thin film solid lubrication in recent years [6,7,8,9,10]. Their applications range from macroscale systems to micro electro-mechanical switches (MEMS) [8]. The frictional properties of deposited graphene-based films have been investigated in many published research articles over the years: Lee et al. [11] used atomic force microscopy (AFM) to reveal the frictional and elastic properties of graphene films and how these properties changed as the number of atomic layers increased. Filleter et al. [12] then built on this research, confirming these findings and indicating that atomic level friction is increased for one- and two-layer graphene, and decreases with an increasing number of layers (until the bulk film can be considered graphitic in nature). Aliyu et al. [13] demonstrated the effectiveness of graphene as a friction reducing additive to a bulk material, by testing a series of ultra-high molecular weight polyethylene (UHMWPE) polymers reinforced with varying loadings of graphene nanoplatelets (GNPs). They found that the addition of graphene to the bulk material reduced the friction experienced compared to bulk UHMWPE under certain operating conditions. Similarly, Xu et al. [14] observed a frictional improvement when adding multilayer graphene to reinforce TiAl matrix composites. Alazemi et al. [15] synthesised a graphene-zinc oxide composite tribological coating and showed that it outperformed the coefficient of friction of a graphene only coating, with a 90% decrease in the wear rate compared to the unlubricated contact. Berman et al. [7] reported a superlubritic condition at the macroscale involving a combination of graphene, nanodiamonds and a diamond-like carbon film (DLC), achieving a coefficient of friction of 0.004. The effect of the environment under which graphene-based coatings operate has also been reported. For example, Bhowmick et al. [16] examined the role of humidity in the tribological performance of graphene films and showed that the lowest coefficient of friction of 0.11 was observed at the highest humidity tested (45% RH).

GO is an oxide derivative of graphene that has similar mechanical and tribological properties [17] and can be synthesised using soft chemistry approaches suitable for large-scale production [18] making it an attractive potential alternative to graphene for applications as industrial tribological coatings.

GO films have shown tribological improvements when used as additives to liquid lubricants [19,20], when used to reinforce a bulk material matrix [13,14] and as thin film solid lubricants [8] such as those examined in this research. It has also been demonstrated that more favourable tribological conditions may be achieved by combining GO with other components to create a composite thin film solid lubricant [15].

One of the challenges in practical application of graphene and GO coatings in industrial applications is employing a suitable coating method to deposit such coating on a range of surface geometries. Whilst some of the simpler methods, such as spin coating, offer low-cost and fast way to apply a variety of coatings, they lack a good control on the thickness and uniformity of the coatings they produce and do not allow for the coating of complex geometries; only flat, radial surfaces [21]. Physical and chemical vapour deposition (PVD and CVD, respectively) techniques offer other, well-explored pathways to achieve high quality and high performance tribological coatings. They also suffer from their own unique drawbacks such as requiring a high vacuum and high cost of operation (in the case of PVD), and a difficulty in coating specific areas (i.e., masking) and high operation temperatures (in the case of CVD) [22]. Electrophoretic deposition (EPD) benefits its high deposition rate, uniformity of deposition, good control of thickness of deposition, ease of scaling-up to industrial quantities and ability to deposit on complex geometries and non-flat surfaces [22,23,24,25,26]. One drawback for EPD coatings is lower adhesion between the substrate and the coating under certain conditions [27].

EPD of GO has been widely explored in previous research, focusing on utilising GO as a corrosion inhibitor [28,29], for consideration in MEMS devices [2], and as macroscale tribological coatings [8].

EPD is selected as the method of coating in this research primarily because, through simple adjustment of the deposition conditions, a range of coatings with varying surface morphologies can be deposited. This will allow for assessment of the tribological impact of such coatings in macroscale applications of complex geometry, such as the piston ring/cylinder liner conjunction within the modern internal combustion engine.

In this context, the aim of the presented research is two-fold: first, to evaluate the effect of deposition conditions on frictional parameters of deposited coatings at the nanoscale asperity level, and second, to develop a novel multi-scale numerical approach for extending the behaviour of these tribological coatings to the macroscale i.e., to realise the conjunction level frictional behaviour of the coatings based on nanoscale measured data.

## 2. Materials and Methods

### 2.1. The Solid Lubricant

GO is prepared using the modified Hummer’s method [30], then separated and centrifuged through a multi-step cleaning process with diluted hydrochloric acid (10% concentration HCl) and de-ionised water (H_2_O), before undergoing drying in a low temperature oven (75 °C) for 12 h. Graphite (C), sulphuric acid (H_2_SO_4_), sodium nitrate (NaNO_3_), potassium permanganate (KMnO_4_) and hydrogen peroxide (H_2_O_2_) used in this method are all purchased from fisher scientific (Loughborough, Leicestershire, UK).

### 2.2. Coating of Substrate

The deposition voltages and deposition times are altered to determine the effect of these conditions on the tribological behaviour of the GO films. The coatings follow a basic naming convention based on the deposition conditions, as displayed in Table 1.

GO is dispersed and sonicated in de-ionised water to create a uniform and stable solution with concentration of 0.1 g/L. Prior to this, AISI 8620 alloy steel substrates with a radius of 4 mm and a height of 4 mm are machined from bar stock and then bead blasted to remove the surface texturing left over from machining, creating surfaces suitable for coating.

The deposition chamber used to conduct electrophoretic depositions has been designed and built specifically to serve this research. The steel substrate acts as the coating electrode and a platinum foil electrode is used as the counter electrode. The distance between the electrodes is maintained at a constant 40 mm for all depositions. Voltage is applied and controlled using a Metrohm PGSTAT204 potentiostat (AutoLab, Kanaalweg, Utrecht, Netherlands).

### 2.3. Metrological Characterisation

A Bruker NPFLEX 3D (Bruker, Coventry, UK) optical profiler with a vertical resolution of 0.15 nm, and an objective magnification of ×20 is used to characterise the coatings by determining the surface characteristics of surface roughness (Sq), peak density (Spd) and peak curvature (Spc). In addition to this, an Alicona InfiniteFocus 3D surface measurement system (Alicona, Graz, Austria) is used to determine the peak height distributions of the deposited films.

### 2.4. Morphological Characterisation

Scanning electron microscopy (SEM, JOEL. Nieuw-Vennep, The Netherlands) is performed using a JEOL 7100 FEGSEM on all deposited coatings and on samples of dried GO flakes before deposition. Electron micrographs from various areas of each coating are captured and interpreted, giving an insight into how the morphology and structure of the deposited coatings is affected by altering the deposition conditions.

Energy-dispersive X-Ray spectroscopy (EDS, Hitachi, Maidenhead, Berkshire, UK) is performed using a Hitachi TM3030Plus Benchtop Scanning Electron Microscope, fitted with an Oxford Instruments Swift ED3000 silicon drift detector (SDD, Oxford Instruments, Abingdon, UK) to gather data about the elemental composition of the coatings. The relative weight % of iron, carbon and oxygen is measured across the coatings. Care is taken to ensure that the selected area is deemed representative of the entire coated substrate.

### 2.5. Tribological Characteristaion

AFM is performed using a Veeco Dimension 3100 AFM (Bruker, Coventry, UK) equipped with Bruker DNP-10 SiN AFM probes (cantilever B) with a stiffness of 0.12 N/m and a nominal tip radius of 20 nm [31]. The shear strength of asperities is found for each coating by utilising AFM in contact mode. Measurements are performed over 5 µm × 5 µm areas across three different sites per coating and then averaged, to provide more reliable data for each coating.

Before each coating is measured, the machine is calibrated using a SiC calibration sample for which the shear strength of asperities is known. A new DNP-10 AFM probe is used to measure each coating, in order to maintain good reliability of measurements. Data is gathered by applying a normal load to the AFM cantilever, measuring the lateral bending of the cantilever as it scans across the surface of the coating, calculating the frictional force at that load, then increasing the applied normal load and measuring the new frictional force at the increased normal load. Nine different normal loads are applied from 1 V to 5 V in increments of 0.5 V, and the resulting load on the cantilever is plotted against the frictional force experienced at the cantilever tip. The gradient of this linear relationship is known as the shear strength of the asperities of the coatings.

### 2.6. Coating Adhesion

Nano-scratch tests are performed on coatings 6.5V1800s, 6.5V3600s and 6.5V5400s using a conical indenter with a radius of 5 µm. Scratches are of total length 600 µm, beginning at a normal load of 0.1 mN for the first 50 µm, then entering a period of progressive loading at a rate of 0.2 mN/µm for 500 µm, before being held at 100 mN for the final 50 µm.

The scratch distance, scratch depth, normal load and lateral frictional force are recorded for all the measured coatings. The scratching frictional force is plotted against the scratch distance, and the area under the curves of those graphs is measured to calculate the work done to remove those coatings. This is used as a rudimentary estimation of the adhesion of the coatings deposited.

### 2.7. Conjunction Level Tribological Analysis

The Greenwood and Tripp contact model [32] is used to predict the conjunction level friction between the two identical AISI 8620 alloy steel surfaces, one coated with GO, and one uncoated. A schematic of this physical model is presented in Figure 1.

This numerical model is valid for nominally flat, rough surfaces with normally distributed peak heights. The peak height distribution of each coating is found using Alicona InfiniteFocus and compared to the Gaussian distribution of the same statistical quantities as the measured data to ensure that the application of this numerical model in this research is valid. This is presented in Section 3.1.

To provide some realistic cases of an industrial application for these coatings, they are considered to be applied at the top dead centre (TDC) of a cylinder liner in an internal combustion engine for the following calculations [33]. Starvation occurs in this region, and higher boundary friction is observed owing to the lack of the formation of a lubricating fluid film.

To calculate the conjunction level friction, f, both boundary friction and viscous friction contributions must be considered (Equation (1)). However, it is assumed that at TDC there is no fluid film because of starvation, so no component of viscous friction, f_v_, which simplifies the subsequent calculations.
f = f_v_ + f_b_(1)

In order to calculate the boundary friction, f_b_, it is necessary to calculate the asperity load, W_a_, and real contact area, A_a_, also (Equation (2)) [34].
f_b_ = τ_0_·A_a_ + 𝜍·W_a_(2)

The real contact area, A_a_, is multiplied by a fixed term, τ_0_, the pressure coefficient. For this research, it is assumed to be 2 MPa [35]. The asperity load is multiplied by the shear strength of asperities, 𝜍, which is determined by measurements from AFM. From Greenwood and Tripp’s research [32], equations are constructed to calculate the asperity load (Equation (3)), W_a_, and the real contact area (Equation (4)), A_a_.
Wa = ((16·√2)/15)·π·(ζ·κ·σ)^2^·√ (σ/κ)·E’·A·F_5/2_(λ)(3)
Aa = π^2^·(ζ·κ·σ)^2^·√ (σ/κ)·A·F_2_(λ)(4)

The peak density, ζ, peak curvature, κ, and surface roughness, σ, are all determined by measurements from optical interferometry, and the statistical functions F_2_(λ) and F_5/2_(λ) are found through previous research [36]. The apparent area of the coating, A, is calculated from the surface area of the substrate, which is simply the area of a circle of radius 4 mm, and the Young’s Modulus of the steel substrate, E’, is obtained from data sheets for AISI 8620 alloy steel [37].

The effect of scale, also widely known as scale effect in literature [38], on the applicability of roughness and friction modelling is a key question in tribology across scales. However, the implemented model here pioneered by Greenwood and Tripp [32], expands the experimentally measured asperity level topography to macro-scale geometry by using statistical approach. At the asperity level, it utilises single asperity contact mechanics based on continuum mechanics. This is a valid assumption, validated experimentally and numerically [39]. The method had been widely used in the literature and validated in different applications such as the valvetrain [35], piston liner [33,40], gears [41], and other macro-scale geometries.

## 3. Results and Discussions

### 3.1. Morphological and Metrological Data

Peak height distribution data is presented for a selection of coatings in Figure 2. The data is overlaid with a Gaussian distribution possessing the same statistical quantities as the measured data to confirm that the distributions have a good agreement with the Gaussian distribution. A Gaussian distribution of peak heights is a requirement for applying the Greenwood and Tripp numerical model, so the validity is confirmed in Figure 2.

The peak height distributions for coatings 6.5V3600s and 6.5V5400s have good agreement with their Gaussian distributions, however coating 6.5V1800s does not exhibit a strong correlation. This data is approximated to the overlaid Gaussian distribution and assumed to be normally distributed in this research.

A Gaussian distribution of peak heights confirms that there exists no underlying discernible patterning (such as texturing the surface) of the coating that could be considered to have an effect on the tribological properties of the surface. The remaining six coatings all exhibit similar correlations with their respective Gaussian distributions as coatings 6.5V3600s and 6.5V5400s.

It is demonstrated in the literature that for electrophoretic depositions, as deposition time increases the thickness of the deposition also increases (with diminishing returns as the deposition inhibits the transfer of charge during the deposition process), and it is also known that for higher deposition voltages, porosity can occur [25]. However, the metrological impact of these factors are less understood. A thicker deposition is expected to perform better as a tribological coating, providing a larger number of shear layers in sliding friction and having a longer wear providing more coating available to wear away. The effect of porosity is not so easily assumed. However, as this may provide a smaller area of contact leading to lower friction, it may also have a large impact on the surface roughness of the deposition depending on the extent of the porosity observed. A selection of metrological characteristics of the depositions are explored with the aim of better understanding the link between the deposition conditions, the physical phenomena they yield and the resulting tribological performance of the deposition.

Metrological data for roughness, density of peaks and arithmetic mean peak curvature for all coatings is presented in Table 2.

Sq refers to the roughness of the coatings, Spd refers to the density of the asperity peaks of the coatings and Spc refers to the curvature of those asperity peaks [42]. The metrological parameters of Sq, Spd and Spc are not the product of solely the deposition conditions. Therefore, specific metrological parameters could be achieved through a number of different combinations of deposition conditions that are not controlled or accounted for in this research. Hence, the metrological parameters of the resulting coatings, in particular the metrological parameters Sq, Spd and Spc which are used in the numerical modelling of the coatings will be referred to in this study.

All three parameters influence the numerical modelling of the boundary friction for each coating. But, they do not change in the same manner with changing deposition conditions. Figure 3 displays the data from Table 2 graphically to better illustrate how these metrological properties of the coatings vary with deposition conditions.

Considering the roughness of the coatings displayed in Figure 3a, it is observed that moderately smoother surfaces are achieved with increasing deposition voltage (except for a large peak for coating 1V3600s, around 1.5 times the roughness of the other coatings). Additionally, varying the deposition times while keeping the deposition voltage constant does not seem to yield a large impact on surface roughness.

Figure 3b shows how the density of peaks varies with deposition conditions. A peculiar trend emerges at the lowest deposition time as the density of peaks decreases with increasing voltage. Then, for a deposition time of 3600 s the density of peaks increases with increasing voltage; the inverse of the trend observed at a deposition time of 1800 s. Finally, at the longest deposition time (5400 s), a slight decrease in density of peaks between the lowest and highest deposition voltage is observed alongside a very low density of peaks for the coating 6.5V 3600s. This data appears to show that the deposition voltage and deposition time have a sizable combined effect on the density of the peaks of the coatings (analogous to the number of points of contact between surfaces), something which has not been reported in relevant literature.

The effect of deposition conditions on the arithmetic mean peak curvature is shown in Figure 3c. Strong variability can be seen across coatings deposited for 1800 s and 3600 s, but a linear trend is observed for coatings deposited for 5400 s. This displays increasing arithmetic mean peak curvature with increasing voltage. This could be attributed to factors such as the pH of the GO solution before deposition [23] and the coating time particularly for longer process.

It is shown that the effect of altering the deposition voltage and deposition time has a measurable and somewhat predictable effect on the achieved metrological parameters of the deposited GO films. This data, when input into the frictional numerical model, enables a better understanding of how the effect of the deposition conditions may be linked to the frictional performance of the coatings at the macroscale.

### 3.2. Mophology and Structure of Depositions

Scanning electron microscopy (SEM) is used to characterize the surface morphology of the deposited coatings and evaluate the impact of deposition conditions on morphological changes.

Images of the coatings deposited at different processing conditions are presented in Figure 4. Results reveal that good homogeneity is achieved for the majority of the coating conditions, although it is less homogeneous in some cases. This will have an impact on the tribological performance of the coatings at the nano-micro scale but will likely not heavily influence the macroscale tribological performance of the coatings, owing to the scale at which the non-homogeneous coatings are observed (hundreds of microns in the case of Figure 4).

An example of the analysis conducted for the coatings in Figure 4 is represented in Figure 5 where the coating 1V1800s has been examined by SEM, EDS and AFM. A low magnification of the coated surface is presented in Figure 5a and EDS spectra for two distinct areas of the surface are depicted in Figure 5b,c; indicating areas of different levels of deposition. The EDS spectra show a strong peak for iron and weak peaks for carbon and oxygen in some areas, indicating a low level of deposition. In other areas, strong peaks for carbon and oxygen are evident, demonstrating that the coating is more predominant.

AFM height maps in Figure 5d–f are taken from three sites on the surface of the coating. It can be seen that surface heights do not differ significantly across the measured areas, and that frictional signal from AFM does not vary significantly over the measured areas. This combination of methods suggests that even though the coating may not be fully homogeneous at some processing conditions (i.e., 1V1800s) it is still possible to gather meaningful and accurate tribological data from these coatings.

Visualisations of the surface of coatings 6.5V1800s, 6.5V3600s and 6.5V5400s obtained from AFM are presented in Figure 6. In Figure 6c, some clearer outlines of ‘platelets’ or ‘flakes’ of coating standing out from the substrate can be observed. 

Figure 6b displays the best coating at the nanoscale, as a good coverage over this area can be seen, with well-defined areas of coating and a low difference between the highest and lowest point measured across the surface.

### 3.3. Tribological Data

The shear strength of asperities for all coatings as determined by AFM is presented in Figure 7.

In order to determine the shear strength of asperities for each coating, three sites are measured and averaged for increased reliability of the data. The data for coating 6.5V1800s is presented in Figure 8. Some of the data points taken at higher loads seem to exhibit a non-linear, almost polynomial response, which is thought to be attributed to the AFM probe undergoing plastic deformation at the higher loads and yielding incorrect results. For this reason, the data points in black are trimmed from the data set and are not used in calculating the average shear strength of asperities.

Considering Figure 7, a trend of decreasing shear strength of asperities can be seen with increasing deposition voltage for deposition times of 3600 s and 5400 s. This trend is not observed for the coatings deposited for 1800 s. In fact, there is a sharp spike in the shear strength of asperities for the coating 6.5V1800s. The conclusion drawn from this is that a deposition time lower than 3600 s is not adequate for the deposition of GO tribological coatings for superior frictional performance. It is also observed that for coatings deposited at the same voltage, a decrease in shear strength of asperities is observed for decreasing deposition time.

### 3.4. Conjunction Level Tribological Analysis

The frictional model for each coating is calculated for Stribeck parameter between 0 and 3. The modelled frictional performance of all coatings is presented in Figure 9a, and the three coatings with the best modelled performance (6.5V1800s, 6.5V3600s and 12V3600s) are isolated in Figure 9b.

From Figure 9a, it can be seen that coatings deposited at 1 V exhibit the highest friction followed by the coatings deposited at 12 V (slightly overlapping with the 6.5 V coatings) and finally the coatings deposited at 6.5 V, which show the lowest friction of all coatings. Interestingly, for all deposition voltages, the coatings deposited for 3600 s possess the lowest friction with changing Stribeck parameter.

### 3.5. Coating Adhesion

The results of nano scratch tests are summarized in Table 3. It is revealed that all coatings are removed during scratching and there is a difference between the work expended to remove them, which is interpreted as the difference in coating adhesion between coatings [43]. As seen in the table below, the work done to remove the coating deposited for 5400 s is greater than that for the coatings deposited at shorter times, although a disposition time of 1800 s yields a greater adhesion than a deposition time of 3600 s. This tentatively suggests that, although the friction of the GO coatings may increase for deposition times below 3600 s, the adhesion may also increase, yielding potentially desirable tribological conditions for some applications requiring a combination of lower friction and increased wear life.

When compared with results in Figure 8, the inverse correlation between boundary friction and coating adhesion is manifested. It is postulated that the coatings with lower shear strength of asperities seen at higher deposition voltages may require less work done to remove these coatings, i.e., poorer surface adhesion, leading to a lower wear life of these coatings.

## 4. Conclusions

Graphene oxide coatings are deposited on steel substrates by electrophoretic deposition with varying deposition voltages and deposition times, creating nine distinct coatings in total. The morphology, structure, adhesion and shear strength of asperities for these coatings are investigated experimentally at the nanoscale asperity level. It is demonstrated that a range of different surface morphologies are created, as seen in the assessment of metrological characteristics (Sq, Spd and Spc). The GO coating with the combination of metrological characteristics most suited for the best nanoscale/microscale frictional performance, as measured by AFM and optical metrological methods, is achieved through the deposition conditions of 6.5 V and 3600 s. The frictional performances of coatings are modelled numerically at conjunction level and correlated with nanoscale measurements. This novel multi-scale approach confirms that the best modelled macroscale frictional performance is achieved by GO coatings with low roughness, low shear strength of asperities and consistent coverage of the steel substrate.

It is found that the surface morphology and nano-scale frictional characteristics have a significant effect on the tribological performance of the resulting GO coatings, as confirmed by modelled frictional performance. Furthermore, it has been shown that the relationship between the deposition time and voltage and the metrological characteristics of the deposited surface may be exploited to enable the tailoring of coatings in order to achieve desired frictional and wear performance. Where previous research has focused on changing the formulation of the coatings and investigating different coating technologies, this research offers another route to achieve potentially desirable tribological coatings by elucidating on the effect of frictional parameters obtained from different deposition conditions. This has the perceived benefit of allowing for greater flexibility and tailoring of tribological coatings deposited by EPD for industrial consideration, without adding additional cost.

Further research exploring a wider range of deposition conditions, such as voltages and times would be valuable in working towards fully realising the tribological potential of GO thin films in industry. Additionally, research into improving the adhesion of these coatings would elevate their potential as realistic industrial solid lubricants.

## Figures and Tables

**Figure 1 materials-13-00041-f001:**
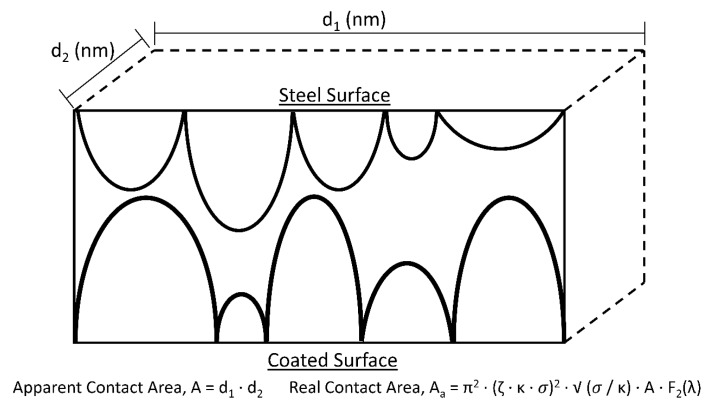
Schematic of physical asperity contact as numerically modelled by Greenwood and Tripp calculations.

**Figure 2 materials-13-00041-f002:**
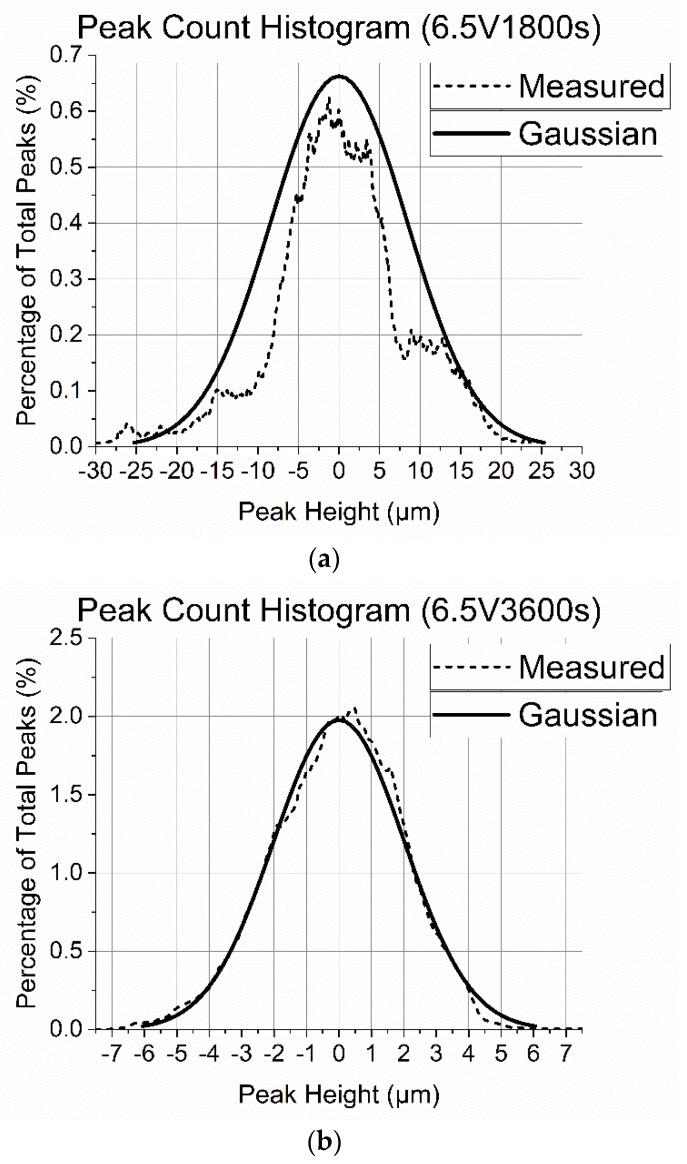
Peak height data for coatings (**a**), 6.5V1800s (**b**), 6.5V3600s and (**c**) 6.5V5400s overlaid with the Gaussian distribution with the same statistical quantities as the measured data.

**Figure 3 materials-13-00041-f003:**
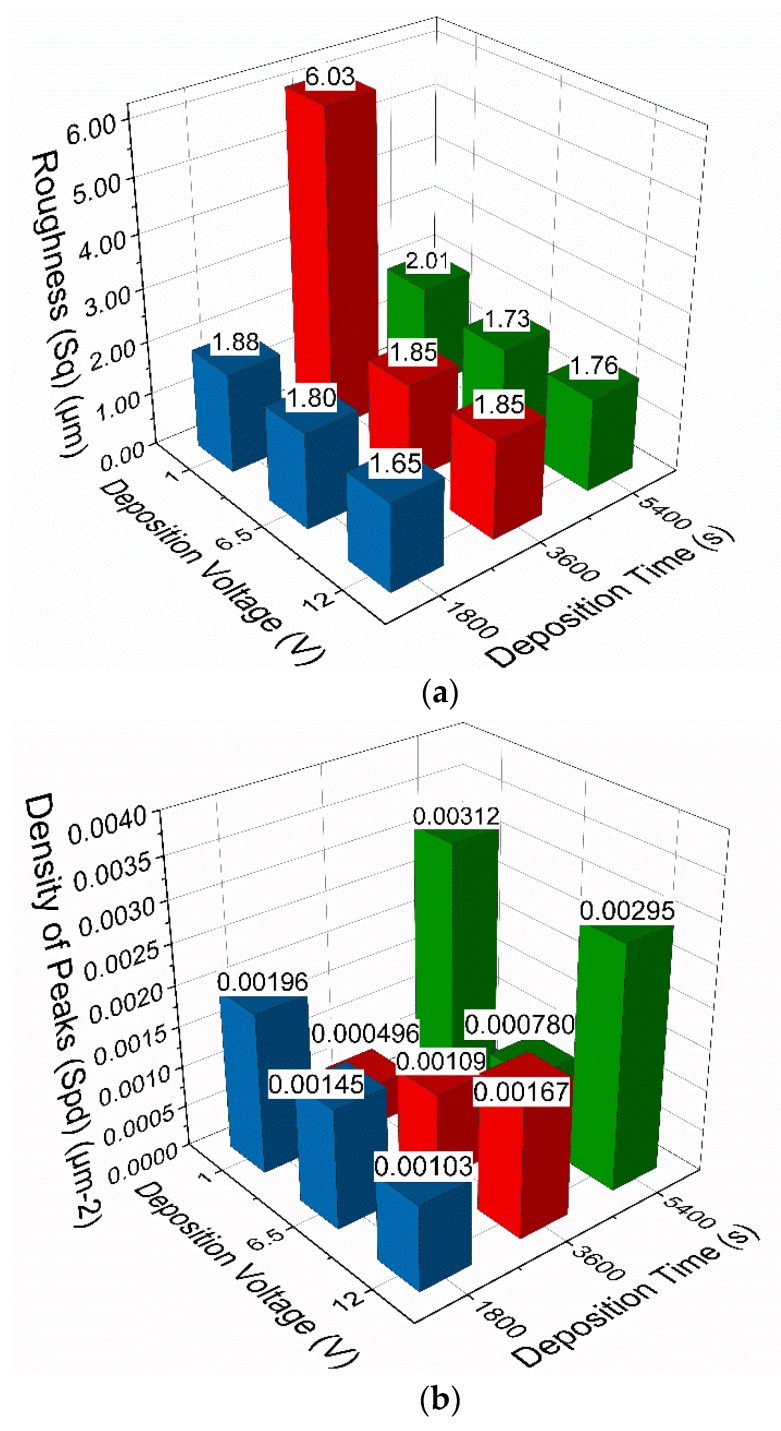
(**a**) Roughness of all coatings; (**b**) density of peaks for all coatings; (**c**) arithmetic mean peak curvature for all coatings.

**Figure 4 materials-13-00041-f004:**
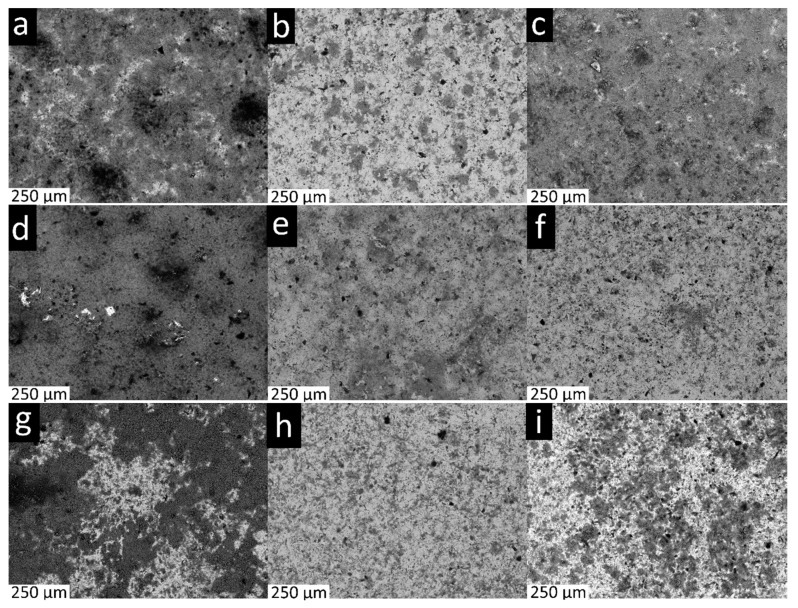
Scanning electron microscopy (SEM) top-down micrographs of coating 1V1800s (**a**), 1V3600s (**b**), 1V5400s (**c**), 6.5V1800s (**d**), 6.5V3600s (**e**), 6.5V5400s (**f**), 12V1800s (**g**), 12V3600s (**h**), 12V5400s (**i**).

**Figure 5 materials-13-00041-f005:**
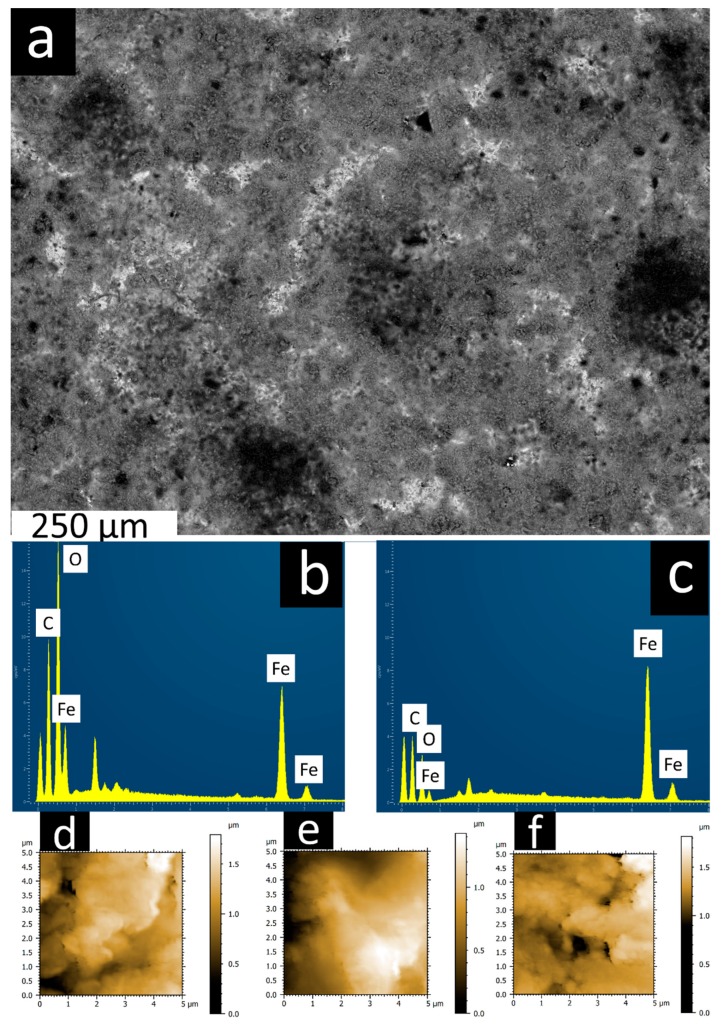
(**a**) SEM image of coating 1V1800s, (**b**) Energy-dispersive X-ray spectroscopy (EDS) spectra of area with good coating, (**c**) EDS spectra of area with bad coating and (**d**–**f**) Height plots of three different area of the coating from atomic force microscopy (AFM).

**Figure 6 materials-13-00041-f006:**
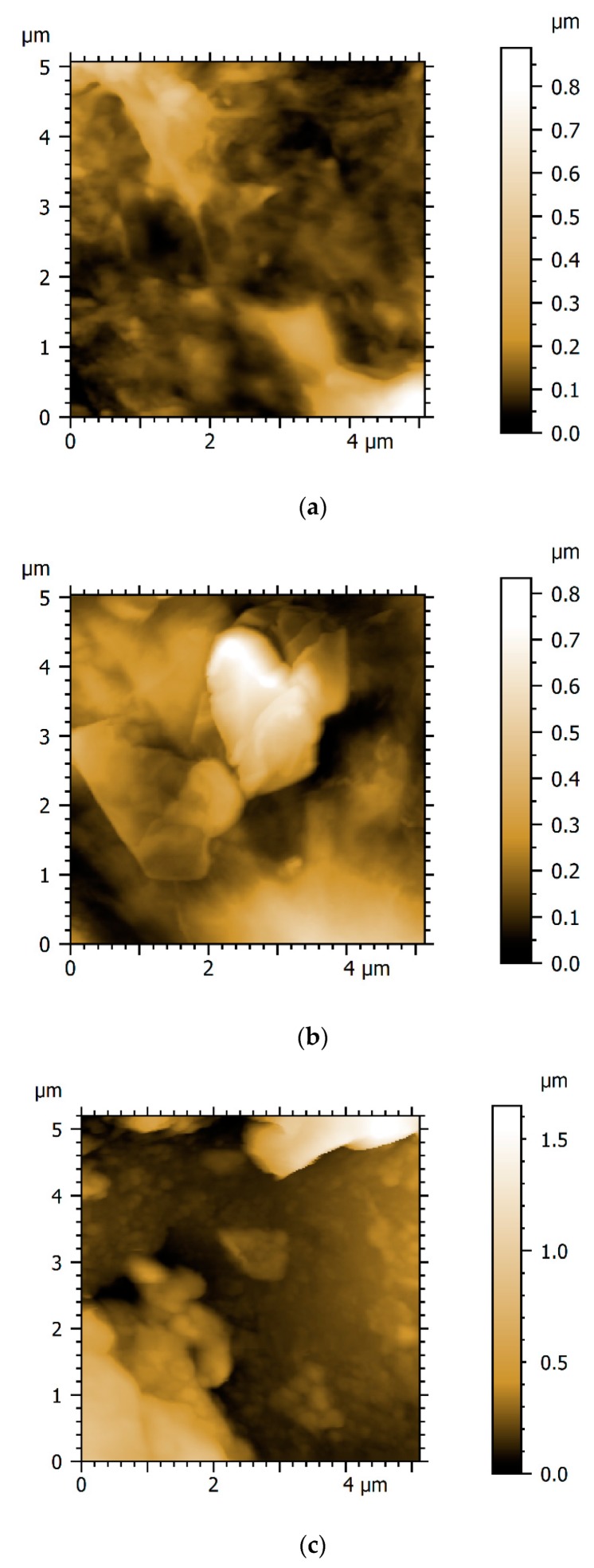
AFM height images from select 5 µm x 5 µm areas on coatings (**a**) 6.5V 1800s, (**b**) 6.5V 3600s and (**c**) 6.5V 5400s.

**Figure 7 materials-13-00041-f007:**
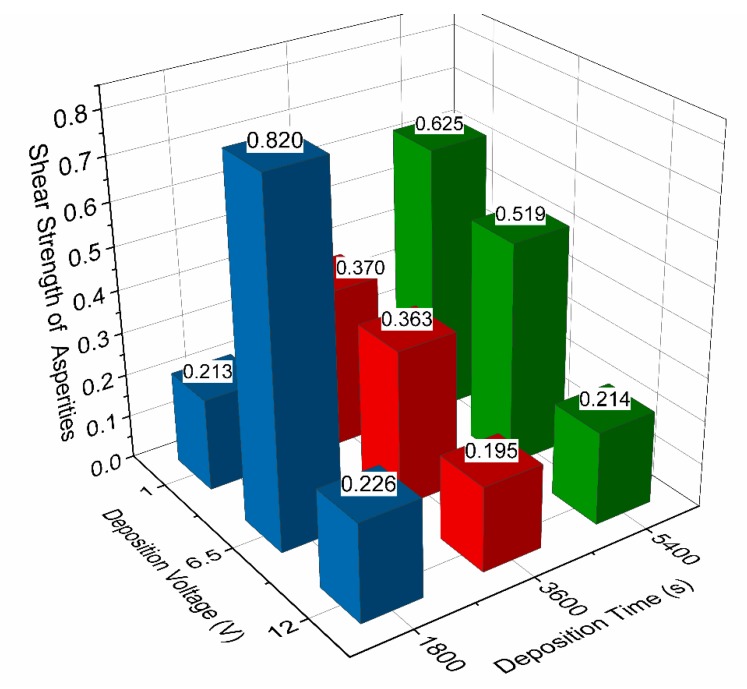
Shear strength of asperities for all coatings from AFM.

**Figure 8 materials-13-00041-f008:**
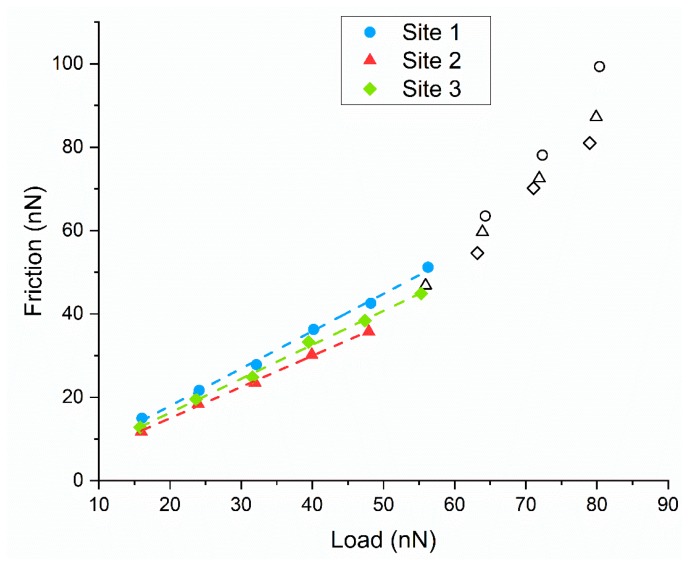
AFM line plots for three sites on coating 6.5V 1800s.

**Figure 9 materials-13-00041-f009:**
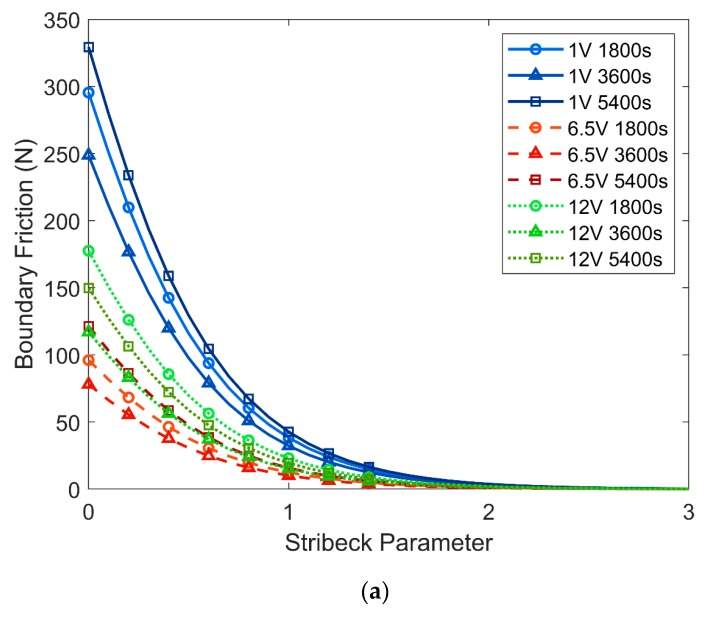
Boundary friction modelled by Greenwood and Tripp calculations for (**a**) all coatings, (**b**) three best performing coatings.

**Table 1 materials-13-00041-t001:** Electrophoretic deposition (EPD) conditions and the associated names of the coatings produced.

Deposition Voltage (V)	Deposition Time (s)	Coating Name
1.0	1800	1V1800s
1.0	3600	1V3600s
1.0	5400	1V5400s
6.5	1800	6.5V1800s
6.5	3600	6.5V3600s
6.5	5400	6.5V5400s
12.0	1800	12V1800s
12.0	3600	12V3600s
12.0	5400	12V5400s

**Table 2 materials-13-00041-t002:** Metrological data for all coatings.

**RMS Height–Sq (µm)**			
	**1 V**	**6.5 V**	**12 V**
**1800 s**	1.88	1.80	1.65
**3600 s**	6.03	1.85	1.85
**5400 s**	2.01	1.73	1.76
**Density of Peaks–Spd (µm^−2^)**			
	**1 V**	**6.5 V**	**12 V**
**1800 s**	0.00196	0.00145	0.00103
**3600 s**	0.000496	0.00109	0.00167
**5400 s**	0.00312	0.00078	0.00295
**Arithmetic Mean Peak Curvature–Spc (µm^−1^)**			
	**1 V**	**6.5 V**	**12 V**
**1800 s**	1.97	1.67	2.27
**3600 s**	2.98	1.58	2.09
**5400 s**	1.74	1.92	2.18

**Table 3 materials-13-00041-t003:** Work done to remove coatings.

Coating Name	Work Done to Remove Coating (µJ)
6.5V 1800s	9.3
6.5V 3600s	9.0
6.5V 5400s	10.3

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
