# Peer review of "Multiscale Numerical and Experimental Analysis of Tribological Performance of GO Coating on Steel Substrates"

_materials, 2019, doi:10.3390/ma13010041_

Round 1

Reviewer 1 Report

The manuscript describes a few properties of GO coated steel substrates in various conditions. 

However, the quality if figures are quite low. It is strongly recommended to improve the quality(eps/ai, font size)of figures, at least figure 5, 6. 

In addition, the benefit and impact of the research should be discussed more in the conclusion. 

Author Response

Response to Reviewers:

Manuscript ID: materials-638661

We are very grateful for the thoroughness and diligence of the learned reviewers and the effort they have expended in dealing with our submission. We have strived to implement all their recommendations. We hope that the amended version of the paper, as well as our point by point responses below meet with our colleagues’ approval. We believe that, due to the comments/suggestions and guidance of our reviewing colleagues, the quality of our paper is significantly improved. We are very grateful.

All the changes made in the revised version of the paper are highlighted for ease of monitoring by our colleagues. Below, we have addressed all the points raised by them individually.

Reviewer 1:

“The manuscript describes a few properties of GO coated steel substrates in various conditions. However, the quality if figures are quite low. It is strongly recommended to improve the quality (eps/ai, font size) of figures, at least figure 5, 6.”
Response: We thank our colleague’s valued feedback on our submission. As stipulated by the MDPI, all figures have been submitted at a resolution of at least 600 DPI separately to the manuscript, although lower quality figures were placed in the manuscript in order to keep the file size down for uploading during the submission process. Additionally, we have made amendments to the figures according to our colleague’s feedback, by increasing the size of text used in figures and re-structuring the presentation of some graphs to improve readability. Specifically regarding figures 5 and 6, based on the recommendation of the other reviewing colleague, they are removed and represented differently in the current version. Below amendments have also been implemented in the revised manuscript.
Figure 1 Increased size of text. De-cluttered presentation.
Figure 2 Increased size of text.
Figure 3 Increased size of text.
Figure 8 Altered presentation for black and white printing.
Figure 9 Increased size of text. “In addition, the benefit and impact of the research should be discussed more in the conclusion.”
Response: Thanks for our learned colleague’s constructive comment. The conclusion section has been re-structured and expanded to discuss the benefits and impacts of the findings in the wider field(s) of tribology and materials science. The changed sections have been highlighted in the revised version.

Reviewer 2 Report

The work mainly reports results with insufficient discussion and analysis and/or explanation of these results. No insight on the relation deposition conditions – structure, morphology of GO coating – tribological properties is provided. Although SEM investigations are mentioned in Sec. 2 “Materials and methods” no results from these investigations are presented. The whole study sounds like a process of optimization without bringing a significant scientific outcome and supported by insufficient data.

Therefore, the paper could not be published in its present form. The content of the paper should be enhanced with additional experimental data and supported by a more profound analysis of relation between technological conditions, structural properties and tribological performance of GO coatings.

Author Response

Response to Reviewers:

Manuscript ID: materials-638661

We are very grateful for the thoroughness and diligence of the learned reviewers and the effort they have expended in dealing with our submission. We have strived to implement all their recommendations. We hope that the amended version of the paper, as well as our point by point responses below meet with our colleagues’ approval. We believe that, due to the comments/suggestions and guidance of our reviewing colleagues, the quality of our paper is significantly improved. We are very grateful.

All the changes made in the revised version of the paper are highlighted for ease of monitoring by our colleagues. Below, we have addressed all the points raised by them individually.

Reviewer 2:

“The work mainly reports results with insufficient discussion and analysis and/or explanation of these results. No insight on the relation deposition conditions – structure, morphology of GO coating – tribological properties is provided.”
Response: Thank you for your valued and insightful feedback on our submission. The discussion, analysis and conclusions in this manuscript is enhanced in line with the comment of our reviewing colleague. To do this, the paper is revised with enhanced analysis, combining analysis from tribological properties, metrological properties, structure, morphology and discussing their relation to deposition conditions. The newly presented analysis of results has been more ‘interlinked’ across deposition conditions, tribological properties, metrological properties, structure and morphology of depositions. Discussion has been made more meaningful by considering all gathered data. The implemented changes are highlighted in sections 3.1, 3.2 and in the conclusions section. “Although SEM investigations are mentioned in Sec. 2 “Materials and methods” no results from these investigations are presented.”
Response: We appreciate this valuable comment of our reviewing colleague. We also admit that this was an oversight during the preparation and submission of the first version of manuscript. We have now included this data, relevant analysis and pertinent discussions in the revised version. SEM images, analysis and relevant discussions have been included in the submission in sections 3.2. “The whole study sounds like a process of optimization without bringing a significant scientific outcome and supported by insufficient data. Therefore, the paper could not be published in its present form.”
Response: This study presents a multi-scale approach for evaluating the tribological performance of thin film solid lubricants and also investigates the effect of deposition conditions on the resulting morphology and tribology of GO films deposited by EPD. The range of deposition conditions presented are not wide-ranging as we are primarily interested in investigating the effect of deposition conditions on the resulting morphology and structure of the deposited films, and not in finding the ‘optimum deposition conditions’ for coatings of this type. The depositions with the best modelled tribological performance are highlighted in an attempt to simplify any comparisons to existing research around this topic. This is now highlighted in the manuscript in section 3.1 and more discussions based on the multi-scale investigation and discovered effect of coating on the tribological performance is provided. “The content of the paper should be enhanced with additional experimental data and supported by a more profound analysis of relation between technological conditions, structural properties and tribological performance of GO coatings.”
Response: Thank you for taking the time to give us this thorough and insightful feedback. Based on your comments, we have revised the content of our submission with additional SEM data and structural, morphological analysis as well as more insightful and meaningful discussion surrounding the relation between the deposition conditions and the structural properties and tribological performance of the GO coatings. SEM data added and analysed alongside structural and morphological analysis.

Reviewer 3 Report

The authors evaluate the friction reducing properties of graphene oxide on steel substrates. The roughness and average shear strength measured in friction testing were used to evaluate the coating properties. While I feel that the paper is somewhat interesting, it does not really give a finding that I find significant. The correlation with deposition time and voltage is very phenomenological and is likely already published in studies unrelated to friction. I also find the graphs confusing, in particular Figure 5. From the other figures, it seems to be difficult to understand the variability in the measured values given. Additionally, the roughness modelling is very interesting, but is also limited because the scaling of the roughness might not match the scaling used in measurements or those relevant in the devices that the authors are looking at. Therefore, I do not think that this manuscript should be published in its current form.

Author Response

Response to Reviewers:

Manuscript ID: materials-638661

We are very grateful for the thoroughness and diligence of the learned reviewers and the effort they have expended in dealing with our submission. We have strived to implement all their recommendations. We hope that the amended version of the paper, as well as our point by point responses below meet with our colleagues’ approval. We believe that, due to the comments/suggestions and guidance of our reviewing colleagues, the quality of our paper is significantly improved. We are very grateful.

All the changes made in the revised version of the paper are highlighted for ease of monitoring by our colleagues. Below, we have addressed all the points raised by them individually.

Reviewer 3:

“The authors evaluate the friction reducing properties of graphene oxide on steel substrates. The roughness and average shear strength measured in friction testing were used to evaluate the coating properties. While I feel that the paper is somewhat interesting, it does not really give a finding that I find significant.”
Response: Thank you for your valued and insightful feedback on our submission. We also find that encouraging to see out colleague’s positive view and interest in our paper. We have now revised our paper with adequate stress on the significance of the presented work. The benefit and impact of our findings are clearly outlined in the introduction and in the revised conclusions section. “The correlation with deposition time and voltage is very phenomenological and is likely already published in studies unrelated to friction.”
Response: We are thankful for this comment of our reviewing colleague. While it is indeed known that as deposition voltage and deposition time increases the thickness of the coated film also increases, this effect on the tribological performance of the films is not so clearly reported in the existing literature. We have now included section detailing expected (known) effects of altering deposition conditions of EPD on resulting films, making clear the missing link between these effects and the resulting changes in ‘tribological effectiveness’ of the films. “I also find the graphs confusing, in particular Figure 5.”
Response: Many thanks for this kind comment. We agree that the findings from EDS are not clear enough in their current form and so have been omitted from the revised version. Instead, we submit one sample EDS spectra in Figure 5 to support the discussion on findings from SEM & AFM in section 3.2 “Morphology and Structure of Depositions. So, Figures 5 & 6 are removed and are replaced by EDS spectra for one coating and discussion alongside SEM & AFM data. “From the other figures, it seems to be difficult to understand the variability in the measured values given.”
Response: The variability in most other figures is noted and the impact this has on the tribology of the coatings is discussed. However, the source of these variabilities and trends were less examined in the previous version. We have now added points of analysis discussing these variabilities and trends and adding clarity to the previously presented graphs. The new sections and discussions are highlighted. “Additionally, the roughness modelling is very interesting, but is also limited because the scaling of the roughness might not match the scaling used in measurements or those relevant in the devices that the authors are looking at. Therefore, I do not think that this manuscript should be published in its current form.”
Response: We thank our reviewing colleague for bringing up this important issue. We are also happy to see our colleague’s interest in this part of our research. The effect of scale, also widely known as scale effect in the literature [1] on the applicability of roughness and friction modelling is a key question in tribology across scales. However, the implemented model here pioneered by Greenwood and Tripp [2], expands the experimentally measured asperity level topography to the macro geometry by using statistical approach. At asperity level, it utilises the single asperity contact mechanics based on continuum mechanics. This is a valid assumption, validated experimentally and numerically [38]. The method had been widely used in the literature and validated in different applications such as valvetrain [3], piston liner [4, 5], gears [6] and other macro geometries. This is now clarified and highlighted in the paper in section 2.7.

[1] Tambe, N. S., & Bhushan, B. (2004). Scale dependence of micro/nano-friction and adhesion of MEMS/NEMS materials, coatings and lubricants. Nanotechnology15(11), 1561.
[2] 31.   Greenwood JA, Tripp JH. The contact of two nominally flat rough surfaces. Proceedings of the institution of mechanical engineers. 1970 Jun;185(1):625-33.
[3] Luan, B., & Robbins, M. O. (2006). Contact of single asperities with varying adhesion: Comparing continuum mechanics to atomistic simulations. Physical Review E74(2), 026111.
[4] Teodorescu, M., Kushwaha, M., Rahnejat, H., & Rothberg, S. J. (2007). Multi-physics analysis of valve train systems: from system level to microscale interactions. Proceedings of the Institution of Mechanical Engineers, Part K: Journal of Multi-body Dynamics221(3), 349-361.
[5] Morris, N., Leighton, M., De la Cruz, M., Rahmani, R., Rahnejat, H., & Howell-Smith, S. (2015). Combined numerical and experimental investigation of the micro-hydrodynamics of chevron-based textured patterns influencing conjunctional friction of sliding contacts. Proceedings of the Institution of Mechanical Engineers, Part J: Journal of Engineering Tribology229(4), 316-335.
[6] Mohammadpour, M., Theodossiades, S., Rahnejat, H., & Dowson, D. (2018). Non-Newtonian mixed thermo-elastohydrodynamics of hypoid gear pairs. Proceedings of the Institution of Mechanical Engineers, Part J: Journal of Engineering Tribology232(9), 1105-1125.

Reviewer 4 Report

In the manuscript “Multiscale Numerical and Experimental Analysis of Tribological Performance of GO Coating on Steel Substrates”, the authors developed a new multi-scale approach based on numerical and experimental methods to investigate the tribological behavior of GO films for potential macroscale applications, such as the piston ring/cylinder liner conjunction within the modern internal combustion engine. The topic of this work is interesting and presents some element of novelty. However, at present, it cannot be published as there are the following weaknesses that must be preventively and adequately addressed:

The english language needs to be revised. There are grammatical and punctuation errors and and inaccuracies in the writing of apexes and subscripts (see units of measurement and chemical formulas). In this work, electrophoretic deposition (EPD) was used to deposit graphene coatings on steel substrates. About this, in the introduction section, the authors are required to explain in more detail, in the light of their results, the advantages/disadvantages of this deposition method compared to other methods reported in the literature (such as chemical vapor deposition (CVD), spin coating, spray deposition, spray drying etc….). This information would give greater importance to the manuscript making it a valid reference for experts in view of future developments on the same subject, while allowing to convince readers more persuasively with regard to the novelty of the content presented in the work. The consideration of these aspects is fundamental also in virtue of the fact that the authors affirm “Although EPD of GO has been widely explored in previous works, focussing on utilising GO as a corrosion inhibitor [24, 25], for consideration in MEMS devices [2], and as macroscale tribological coatings [8], the effect of the coating method and deposition parameters on the tribological performance of GO films has not been investigated before. “ In which mode (contact, tapping ...) were AFM images acquired? This missing information should be added to the experimental part. The topographical characterization by SEM and EDX is lacking, although the authors describe the aforementioned techniques in the experimental part, stating: "Electron micrographs from various areas of each coating were captured and interpreted, giving an insight into how the morphology and structure of the deposited coatings is affected by altering the deposition parameters.” In this regard, the authors are asked to add SEM-EDX micrographs in the manuscript and discuss them.

Author Response

Response to Reviewers:

Manuscript ID: materials-638661

We are very grateful for the thoroughness and diligence of the learned reviewers and the effort they have expended in dealing with our submission. We have strived to implement all their recommendations. We hope that the amended version of the paper, as well as our point by point responses below meet with our colleagues’ approval. We believe that, due to the comments/suggestions and guidance of our reviewing colleagues, the quality of our paper is significantly improved. We are very grateful.

All the changes made in the revised version of the paper are highlighted for ease of monitoring by our colleagues. Below, we have addressed all the points raised by them individually.

Reviewer 4:

“The english language needs to be revised. There are grammatical and punctuation errors and and inaccuracies in the writing of apexes and subscripts (see units of measurement and chemical formulas).”
Response: Thank you for your valued and insightful feedback on our submission. Further to this comment, we have gone through and corrected the inaccuracies in the writing of apexes and subscripts and have improved the grammar used throughout the submission. “In this work, electrophoretic deposition (EPD) was used to deposit graphene coatings on steel substrates. About this, in the introduction section, the authors are required to explain in more detail, in the light of their results, the advantages/disadvantages of this deposition method compared to other methods reported in the literature (such as chemical vapor deposition (CVD), spin coating, spray deposition, spray drying etc….). This information would give greater importance to the manuscript making it a valid reference for experts in view of future developments on the same subject, while allowing to convince readers more persuasively with regard to the novelty of the content presented in the work. The consideration of these aspects is fundamental also in virtue of the fact that the authors affirm “Although EPD of GO has been widely explored in previous works, focussing on utilising GO as a corrosion inhibitor [24, 25], for consideration in MEMS devices [2], and as macroscale tribological coatings [8], the effect of the coating method and deposition parameters on the tribological performance of GO films has not been investigated before. “”
Response: We are thankful for this clarifying comment. The choice to use EPD has not been adequately explained before; and there are indeed good reasons for using this method over other available industrial methods. We recognise that explaining the advantages/disadvantages of using this method in more detail as well as providing a critical comparison of coating methods would greatly convince the reader of the significance of our submission as well as giving the manuscript greater importance, making it a valid reference. A section is now added, critically analysing currently available coating methods and clarifying reasons of the choice to use EPD with reference to novelty of the work presented. This new section is now highlighted. “In which mode (contact, tapping ...) were AFM images acquired? This missing information should be added to the experimental part.”
Response: AFM was used in contact mode for this research. This omission has been rectified within the text for resubmission in section 2.5. “The topographical characterization by SEM and EDX is lacking, although the authors describe the aforementioned techniques in the experimental part, stating: "Electron micrographs from various areas of each coating were captured and interpreted, giving an insight into how the morphology and structure of the deposited coatings is affected by altering the deposition parameters.” In this regard, the authors are asked to add SEM-EDX micrographs in the manuscript and discuss them.”
Response: Many thanks for bringing this important point to our attention. The SEM data combined with discussions around the morphology and structure of the depositions and along with data gathered from EDS are now included in the main text.

Round 2

Reviewer 2 Report

The authors have taken into consideration all the reviewer's remarks and suggestions. The paper is significantly improved and can be published in the present form.

Author Response

Many thanks for your kind feedback on our manuscript.

Reviewer 4 Report

The authors have sufficiently addressed the issues raised by the reviewer.

Therefore, the revised manuscript can be accepted for publication.

Author Response

Many thanks for your kind feedback on out manuscript.